# The feasibility and clinical significance of lateral approach thyroidectomy

Ran An[1,2☯], Yong-Xue Gu[2☯], Xi-Hao Ni[1], Ying Lei[2], Wei-Tao Wang[1], Xiao-Juan Men[2], Jing-Yi Ma[1], Chang-Liang Wang[2]*

1 School of Clinical Medicine, Shandong Second Medical University, Weifang, Shandong Province, China,
2 Department of Thyroid and Breast Surgery, Weifang People's Hospital, Weifang, Shandong Province, China

☯ These authors contributed equally to this work.
* wfwangchangl@sina.com

**Data Availability Statement:** All relevant data are within the manuscript and its Supporting Information files.

**Funding:** This work was supported by Weifang Health and Family Planning Commission

## Abstract

### Background

By comparing the three lateral approaches to thyroidectomy, the feasibility and clinical effects were analyzed, and the advantages of the lateral approach were summarized.

### Methods

From January 2022 to January 2023, 52 patients with thyroid cancer admitted to our department were selected and subjected to Lateral approach for thyroidectomy. Among them, 31 patients underwent thyroidectomy via the supraclavicular approach, 13 patients underwent endoscopic thyroidectomy via the subclavicular approach, and 8 patients underwent endoscopic thyroidectomy via the axillary approach. The basic conditions, surgical conditions, complications, postoperative pain scores and postoperative satisfaction of patients in the three approach surgery groups were recorded and analyzed.

### Results

There were no significant differences among the three approach groups in terms of patient characteristics, number of central lymph node dissections, intraoperative blood loss, postoperative drainage volume, duration of drainage tube placement, length of hospital stay, postoperative pain, satisfaction, and complications. However, the operation time was longest in the subclavicular approach group, followed by the axillary approach group, and shortest in the supraclavicular approach group. The total hospitalization cost was highest in the axillary approach group, followed by the subclavicular approach group, and lowest in the supraclavicular approach group.

### Conclusion

The lateral approach for thyroidectomy is deemed a safe and effective method. The three different approach paths gradually increase in length, allowing for the accumulation of anatomical experience. This approach has a shorter learning curve for clinical doctors and is a favorable choice for patients seeking aesthetic benefits.

(wfwsjs_2018_029). The funders had no role in study design, data collection and analysis, decision to publish, or preparation of the manuscript.

**Competing interests:** The authors declare no conflicts of interest.

# 1 Introduction

As people increasingly prioritize health checkups and thyroid ultrasound technology becomes more widespread, the detection rate of thyroid cancer continues to rise. Thyroid cancer is a prevalent malignancy primarily treated through surgery [1]. In recent years, thyroid cancer has been more commonly diagnosed in younger female patients, who often place a higher value on aesthetic outcomes [2]. Traditional neck incisions can impact appearance, particularly in patients with a predisposition to scarring. For some young, unmarried patients, this can create significant psychological burden. In addition to traditional open surgery, current surgical approaches include transoral, submental, retroauricular, supraclavicular, subclavicular, axillary, and breast approaches [3]. This study examines the lateral approach for thyroid surgery, which encompasses open surgery through an oblique incision above the clavicle and endoscopic surgery through the subclavicular and axillary approaches. The introduction of endoscopic technology has enabled more concealed incisions and precise surgery. This approach is conducive to central lymph node dissection, as well as technical challenges such as protecting the recurrent laryngeal nerve and parathyroid glands. However, it is only suitable for unilateral thyroid surgery [4, 5]. The lateral approach is associated with minimal trauma, and the approach path gradually lengthens from the supraclavicular to the subclavicular and axillary (Fig 1), resulting in a progressive accumulation of anatomical experience. As such, this approach has a shorter learning curve for clinical doctors and may increase the success rate for beginners. This study aims to evaluate and summarize the advantages of the lateral approach by comparing the feasibility and clinical outcomes of three different approaches for thyroid surgery, with the ultimate goal of maximizing patient benefits.

# 2 Materials and methods

## 2.1 General information

Fifty-two patients diagnosed with thyroid cancer who were admitted to Weifang People's Hospital between January 2022 and January 2023 were included in this study. The study has been approved by the Ethics Committee of Weifang Medical University (Approval No.

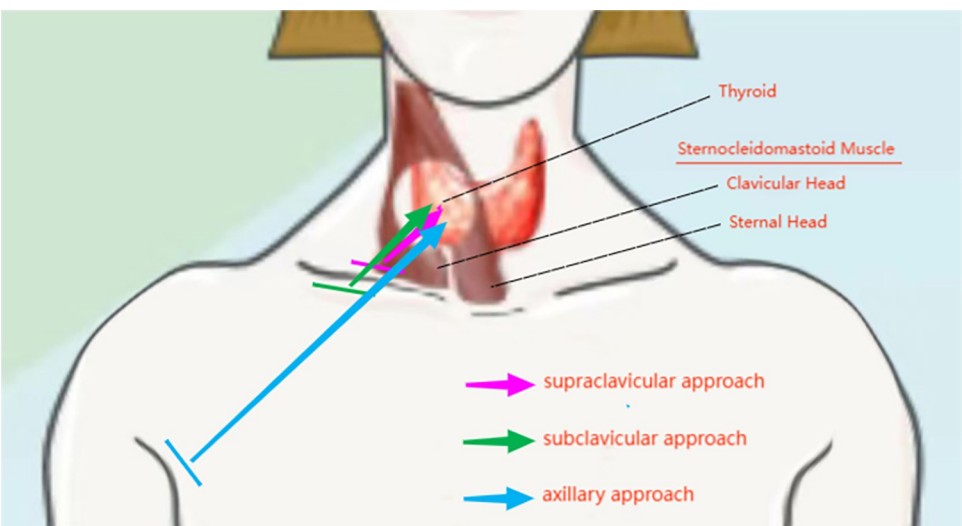

**Fig 1. The lateral approach progresses from the supraclavicular region to the subclavicular region and further to the axilla, passing through the natural gap between the clavicular head and sternal head of sternocleidomastoid muscle, with the pathway gradually increasing in length.**

2023YX134). Among them, 31 underwent thyroidectomy via the supraclavicular approach, 13 underwent endoscopic thyroidectomy via the subclavicular approach, and 8 underwent endoscopic thyroidectomy via the axillary approach. Of the 52 patients, 8 were male and 44 were female, with an age range of 31–50 years. Inclusion criteria were: absence of neck radiation history; ultrasound indicating a nodule located within a single lobe of the thyroid gland without invasion of the thyroid gland; no significant abnormalities in preoperative blood routine and coagulation routine; fine-needle aspiration biopsy indicating papillary thyroid carcinoma, and imaging not showing lymph node metastasis in the neck region; no serious cardiovascular and cerebrovascular diseases, and tolerable cardiac and pulmonary function; and signed informed consent from all patients. Exclusion criteria were: the presence of evidence of lateral neck lymph node metastasis based on needle biopsy; ultrasound indications of bilateral thyroid nodules requiring total thyroidectomy; presence of other malignant tumors; abnormal coagulation function; serious cardiovascular and cerebrovascular diseases; and inability to cooperate with treatment.

## 2.2 Methods

**2.2.1 Surgical equipment.**　The surgical equipment used in this study included a high-definition laparoscopic display system (OLYMPUS EVIS EXERA III CV-190), an ultrasonic scalpel (Johnson & Johnson ETHICON ENDO-SURGERY GEN11 Generator), a high-frequency electric scalpel (ERBE VIO300S), and a recurrent laryngeal nerve monitoring system (inomed Medizintechnik GmbH C2 NerveMonitor). Instruments used in laparoscopic procedures include an electrocoagulation hook, a suspension hook, non-traumatic grasping forceps, separation forceps, tissue scissors, needle holders, vessel sealing devices, specimen retrieval bags, and others, provided by Hangzhou Tonglu Yida Medical Appliance And Equipment CO.LTD.

**2.2.2 Surgical technique.**　All patients underwent endotracheal intubation under general anesthesia and were placed in a supine position with a pillow under their shoulders and their head tilted back. For the subclavicular and axillary endoscopic approaches, the patient's head was slightly turned to the opposite side, while for the axillary approach, the ipsilateral upper limb was externally rotated by 90˚.

1. Supraclavicular approach: At 1 cm above the clavicle, a curved incision approximately 5 cm in length was made. Along the deep surface of the platysma muscle, a skin flap was dissected, extending from the thyroid cartilage above to the upper margin of the sternum below. Using a thyroid retractor, the skin flap was pulled open. The dissection extended between the sternal and clavicular heads of the sternocleidomastoid muscle, opening the strap muscles, and exposing the thyroid for thyroid surgery (including pre-tracheal and central compartment lymph node dissection).

2. Subclavicular approach: A 4 cm incision was made below the clavicle, and a trocar was inserted for endoscopic visualization. The subcutaneous tissue was separated, and the sternocleidomastoid muscle was exposed using an ultrasonic scalpel. The natural gap between the clavicular head and sternal head of sternocleidomastoid muscle was explored and separated, and the space was enlarged using a hook. The dissection is performed between the internal jugular vein and the lateral border of the sternocleidomastoid muscle, with the strap muscle freed on its deep surface, exposing the thyroid for thyroid surgery.

3. Gasless transaxillary approach: A 4 cm incision was made at the second axillary fold, and a trocar was inserted at the intersection of the anterior axillary line and the edge of the breast. The thyroid gland was exposed for surgery under endoscopic visualization, and the surgical

steps were the same as those for the subclavicular approach. A special suspension hook was used to assist in the exposure of the space.

During surgery, meticulous dissection was performed to protect the parathyroid glands and recurrent laryngeal nerves, and hemostasis was carefully performed. The surgical field was irrigated, a drainage tube was placed at the incision site, and negative pressure drainage was applied. The incision was closed using 4–0 absorbable sutures.

## 2.3 Observational indicators

2.3.1 Basic information such as age, gender, body mass index, tumor size, and number of dissected lymph nodes were collected for the three groups of patients.

2.3.2 The surgical situations were observed and compared among the three groups, including the operative time, intraoperative blood loss, postoperative drainage volume, duration of drainage tube placement, length of hospital stay, and total hospitalization costs.

2.3.3 The incidence of complications was observed and compared among the three groups, including transient hoarseness, transient hypoparathyroidism, dysphagia, bleeding, infection, and chyle leakage.

2.3.4 Postoperative pain was evaluated using the visual analogue scale (VAS), with a total score of 10 points and a lower score indicating less pain. Pain scores were recorded on the first and third postoperative days.

2.3.5 Patients were followed up for three months after surgery to evaluate their satisfaction, including swallowing function, skin sensation, and incisional appearance, based on subjective evaluation using a 5-point scale: 1 (very satisfied), 2 (satisfied), 3 (fair), 4 (dissatisfied), and 5 (very dissatisfied).

## 2.4 Statistical analysis

Statistical analysis was performed using IBM SPSS Statistics 25 software. Box plots were generated using GraphPad Prism 8. Continuous data were presented as mean ± standard deviation ($\bar{x}\pm s$). Analysis of variance (ANOVA) was used for normally distributed data, and nonparametric tests were used for data with non-uniform variances. Categorical data were presented as frequency (n) and percentage (%), and the chi-square test was used for intergroup comparisons. A p-value of less than 0.05 was considered statistically significant.

## 3 Results

All patients in the three surgical groups underwent the operations successfully.

### 3.1 Comparison of patients baseline characteristics

This study included a total of 52 patients, comprising 8 males and 44 females, with ages ranging from 31 to 50 years and a mean age of (41.75±9.27) years. No patient in any group was obese or underweight. There were no statistically significant differences in age, body mass index, or tumor size among the three surgical groups (P>0.05). Moreover, the number of central lymph nodes removed was also comparable among the groups (P>0.05), indicating that the efficacy of central lymph node dissection was similar for all three approaches. See Table 1 for details.

### 3.2 Comparison of patients' surgical situations

The operation time was defined as the duration from skin incision to closure, which was significantly longer in the group undergoing the subclavicular approach, followed by the axillary

**Table 1. Comparison of baseline characteristics of patients in three surgical approach groups ('x±s).**

| Variables | Supraclavicular | Subclavicular | Axillary | F value | P value |
|---|---|---|---|---|---|
| Sex (Male/Female) | 6/25 | 2/11 | 0/8 | - | - |
| Age (years) | 40.42±8.84 | 43.62±7.68 | 43.88±13.05 | 0.786 | 0.461 |
| BMI (kg/m2) | 23.18±2.81 | 24.07±2.72 | 23.64±2.18 | 0.516 | 0.600 |
| Tumor size (mm) | 11.39±12.4 | 8.41±9.95 | 18.38±11.86 | 1.799 | 0.176 |
| Number of lymph nodes | 3.71±1.19 | 4.23±0.83 | 3.50±1.31 | 1.322 | 0.276 |

approach, and the supraclavicular approach had the shortest duration (P<0.05). Minimal bleeding occurred in all three groups during the surgery, and there were no significant differences in the amount of bleeding, drainage volume, duration of drainage tube placement, or length of hospital stay (P>0.05). The total hospitalization cost was highest in the axillary approach group, followed by the subclavicular approach group, and lowest in the supraclavicular approach group, with statistically significant differences (P<0.05). See Table 2 for details. Additionally, we have graphed statistically significant box plots for operation time and total hospitalization cost, as detailed in Fig 2. The box plots distinctly illustrate the previously mentioned findings. However, it is essential to note that our sample size is relatively small, which may introduce uncertainty into the results.

### 3.3 Comparison of postoperative complications

No patients in any group experienced wound bleeding (including incision oozing or hematoma), wound infection, or chyle leakage. Only one patient in the group with the supraclavicular approach above the clavicle experienced transient hoarseness, which was treated with nutritional nerve therapy and gradually recovered within one month of follow-up. There were no statistically significant differences among the three groups (P>0.05). Postoperative transient hypoparathyroidism may occur, including varying degrees of lip and hand numbness, which can be restored to normal by administering calcium gluconate. There were no statistically significant differences among the groups (P>0.05). The incidence of recurrent laryngeal nerve injury and hypoparathyroidism was very low, possibly due to meticulous dissection. Other thyroid surgery-related complications did not occur. See Table 3 for details.

### 3.4 Comparison of pain scores at different time points

The incisional pain was significant in all three groups on the first day after surgery, but was basically relieved by the third day after surgery without severe pain. The VAS score decreased in all groups, and there was no statistically significant difference (P>0.05). See Table 4 for details.

**Table 2. Presents the comparison of surgical parameters among the three surgical approach groups ('x±s).**

| Variables | Supraclavicular | Subclavicular | Axillary | F/$x^2$ value | P value |
|---|---|---|---|---|---|
| Operation time(min) | 86.06±16.76 | 171.77±45.78 | 156.00±38.74 | 27.211 | <0.001 |
| Blood loss(ml) | 12.26±5.95 | 11.62±1.26 | 11.38±1.06 | 0.158 | 0.854 |
| Day 1 drainage volume(ml) | 41.29±20.45 | 51.92±14.22 | 51.88±29.39 | 1.644 | 0.204 |
| Total drainage volume(ml) | 53.94±24.46 | 72.85±15.10 | 73.13±48.25 | 3.000 | 0.059 |
| Duration of drainage tube placement(d) | 2.68±0.48 | 2.69±0.48 | 2.75±0.46 | 0.074 | 0.928 |
| Length of hospital stay(d) | 6.52±1.03 | 6.85±1.07 | 7.25±1.28 | 1.609 | 0.210 |
| Total hospitalization cost(CNY) | 18431.71±3044.48 | 23839.23±905.11 | 24170.13±1089.93 | 36.659 | <0.001 |

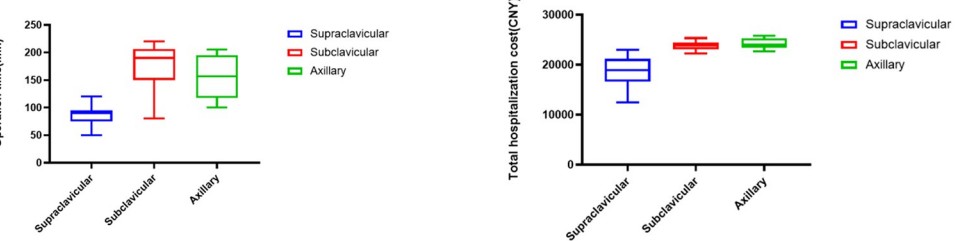

**Fig 2. Box plots for operation time and total hospitalization cost among three surgical approach groups.**

## 3.5 Comparison of patient satisfaction at 3 months postoperatively

After a 3-month follow-up, including evaluation of swallowing function, skin sensation, and incision appearance, the overall satisfaction rate was 100% in all three groups of patients who underwent surgery via the supraclavicular, infraclavicular, and axillary approaches, and the differences were not statistically significant (P>0.05). See Table 5 for details.

## 4 Discussion

Thyroid surgery has always been a concern for patients, with the appearance of the incision becoming a focus of attention for thyroid surgeons. Technological advances have led to the development of various surgical approaches, including lateral approaches and those through artificial cavities such as the mouth, chest, and breast, to achieve better cosmetic results. Currently, common hidden incision surgical approaches include oral, submental, retroauricular, supraclavicular, subclavicular, axillary, and breast routes [3]. Endoscopic thyroid surgery, which has been developed over the past 20 years, has been recognized for its safety and feasibility, with undeniable cosmetic benefits. The magnification effect of the endoscope during surgery allows for clearer and more precise identification and manipulation of the parathyroid and recurrent laryngeal nerves. This, coupled with meticulous dissection techniques, results in precise operations, leading to low complication rates. However, each surgical approach has its advantages and disadvantages in terms of safety, cosmetic results, and trauma [3, 6]. Although hidden incision approaches conceal the incision, they may result in increased endoscopic surgery time and hospitalization costs, and require higher proficiency in surgeons and instrument requirements [7]. Lateral approaches are only suitable for unilateral lesions [5]. In addition, the carbon dioxide required for insufflation may have adverse effects on hemodynamics and blood gas levels, such as hypercapnia, acidosis, and decreased mean arterial and central venous pressure [8]. Oral routes have risks of facial paralysis, lip movement disorders, etc., and change the incision from a class I to a class II incision, increasing the risk of infection in patients [9].

**Table 3. Comparison of complications among the three approaches ('x±s).**

| Variables | Supraclavicular(n = 31) | Subclavicular(n = 13) | Axillary(n = 8) | $x^2$ value | P value |
|---|---|---|---|---|---|
| hoarseness (transient) | 1 (3.2%) | 0(0.0%) | 0(0.0%) | 0.691 | 0.708 |
| Hypoparathyroidism (transient) | 2 (6.5%) | 0(0.0%) | 0(0.0%) | 1.409 | 0.494 |
| Water choking | 0 | 0 | 0 | - | - |
| Bleeding | 0 | 0 | 0 | - | - |
| Infection | 0 | 0 | 0 | - | - |
| Chyle leakage | 0 | 0 | 0 | - | - |
| Others | 0 | 0 | 0 | - | - |

**Table 4. Incision pain score at different time points after surgery in the three groups ('x̄±s).**

| Variables | Supraclavicular | Subclavicular | Axillary | F value | P value |
|---|---|---|---|---|---|
| Postoperative day 1 | 2.58±0.50 | 2.54±0.52 | 2.63±0.52 | 0.074 | 0.929 |
| Postoperative day 3 | 1.00 | 1.00 | 1.00 | - | - |

Open surgery, as a classic surgical technique, is intuitive, concise, and thorough, and is suitable for all thyroid surgeries. At present, there is no new surgical technique that can replace its role and status [10].

The lateral approach to thyroid surgery includes open surgery through an oblique incision above the clavicle and endoscopic surgery through the subclavicular and axillary approaches. Studies have shown that for patients who require a second operation due to tumor recurrence, the lateral approach is superior to the traditional anterior approach in terms of surgical time, blood loss, and complications, as it avoids the disadvantages of the latter approach caused by the disorder of normal anatomical structures and scar tissue proliferation [11]. However, this approach is only suitable for unilateral thyroid surgery, and for patients with abundant neck fat or developed muscles, the exposure space for surgical operations is relatively poor, increasing the difficulty of the operation [5]. ①The open thyroidectomy via a oblique incision above the clavicle inherits the advantages of the simple and easy-to-learn open surgery, while ensuring the beauty of the incision by reducing the probability of scar tissue proliferation, since the skin tension above the clavicular fossa is smaller and the incision line is consistent with the skin texture [12]. Also, since the incision is deviated from the anterior neck, postoperative clothing or necklaces can cover the incision, achieving a certain degree of aesthetic requirement. ②The endoscopic thyroidectomy via an incision below the clavicle also follows the skin texture and uses the natural cavity to create a small surgical cavity, significantly reducing damage [13]. It also has minimal interference with the appearance of the neck, and does not require incision of the neck white line. After the surgery is completed, all anatomical structures can naturally return to their original positions, effectively protecting the patient's neck sensation, reducing postoperative discomfort in swallowing, and avoiding neck-swallowing linkage [14]. In addition, the surgical incision is moved from the exposed neck to the covered clavicular area, concealing the wound and meeting the aesthetic needs of the patients. ③The transaxillary approach endoscopic thyroidectomy was initially described by Japanese scholar Ikeda in 2000 [15]. Subsequently, Yoon reported the first gasless transaxillary approach endoscopic thyroid surgery in 2006 [16]. Over the course of more than a decade, this surgical technique has undergone development and refinement, gradually earning recognition among professionals in the field. The incision is made in the armpit, resulting in no visible scars on the neck skin. Moreover, due to the surgical pathway, the patient's anterior neck function area is effectively

**Table 5. Comparison of patient satisfaction among three surgical approaches at 3 months postoperatively[n (%)].**

| Variables | Supraclavicular (n = 31) | Subclavicular(n = 13) | Axillary(n = 8) | $x^2$ value | P value |
|---|---|---|---|---|---|
| Very satisfied (1 points) | 29(93.5%) | 13(100%) | 8(100%) | 1.409 | 0.494 |
| Satisfied (2 points) | 2 (6.5%) | 0(0.0%) | 0(0.0%) | | |
| Fair (3 points) | 0 | 0 | 0 | - | - |
| Unsatisfied (4 points) | 0 | 0 | 0 | - | - |
| Very unsatisfied (5 points) | 0 | 0 | 0 | - | - |
| Overall satisfaction (≤3 points) | 31(100%) | 13(100%) | 8(100%) | - | - |

Note: Overall satisfaction was evaluated based on the total score, with a score of 1–3 indicating satisfaction and a score of 4–5 indicating dissatisfaction.

protected. This approach ensures comprehensive surgery while simultaneously offering both cosmetic and functional benefits for individuals with thyroid disorders [17].

## 4.1 Analysis of the number of central lymph nodes in the cleaning process

The lateral approach provides an inherent advantage in exposing the recurrent laryngeal nerve and anatomical structures such as the trachea and carotid artery [4]. However, due to the obstruction caused by the clavicle, the lower limit of the cleaning process can only reach the level of the upper edge of the innominate artery. Studies have shown that this range of cleaning is completely sufficient for N0 thyroid cancer patients, while for patients with suspicious lymph nodes below the innominate artery or in deeper areas, it is recommended to use the anterior approach [18]. According to the results of this study, there was no significant difference in the number of central lymph nodes cleaned among the three approaches, indicating that the effectiveness of cleaning the central lymph nodes was similar among the three surgical groups. As for their long-term recurrence rate, it is not yet clear due to the short follow-up period.

## 4.2 Analysis of operation time

Our study has revealed that the subclavicular approach took the longest time, followed by the axillary approach, and the supraclavicular approach was the shortest. Firstly, the use of two endoscopic approaches took more time compared to the supraclavicular approach. This was possibly due to the longer path and the need for separation. Moreover, endoscopic surgery is usually carried out by a single operator, which may increase surgical time due to limitations in assistance and lens cleaning. Secondly, the subclavicular approach took slightly longer than the axillary approach. This was likely due to the experience and proficiency accumulated through previous supraclavicular and subclavicular approach surgeries, laying the foundation for the axillary approach. Endoscopic thyroidectomy necessitates the surgeon to possess a strong proficiency in endoscopic techniques and a thorough knowledge of anatomy. The experience gained through the supraclavicular approach serves as a fundamental basis for the subclavicular and axillary approaches. In essence, the subclavicular and axillary approaches can be considered as extensions of the supraclavicular approach. Hence, the surgical techniques are interconnected, resulting in a shorter learning curve. The transoral approach requires the operator to adapt to and change their spatial perception due to the opposite surgical orientation compared to open surgery [9]. The trans-mammary approach requires the establishment of a longer tunnel in the chest and neck, which makes central lymph node dissection relatively difficult [19]. Therefore, we believe that the lateral approach has certain advantages for both physicians and patients.

## 4.3 Complication analysis

In our study, only one case of transient hoarseness occurred in the supraclavicular approach group, which was successfully managed with conservative treatment of the recurrent laryngeal nerve. The patient gradually recovered within one month of follow-up. The possible reasons for this complication may include thermal injury to the recurrent laryngeal nerve during ultrasonic scalpel use, traction-induced stimulation during surgery, and transient compression of the nerve due to postoperative edema of the surrounding tissues [20]. In addition, a few patients experienced transient symptoms of hypoparathyroidism, which were resolved with calcium and vitamin D supplementation. Although the postoperative symptoms caused by hypoparathyroidism were mild and short-term, it is important to pay close attention to them to prevent the potential risk of tetany [21]. In addition, there is an increased risk of vocal cord

paralysis [22]. No severe complications occurred after surgery, indicating that the lateral approach for thyroid surgery is safe.

### 4.4 Satisfaction analysis

As thyroid cancer becomes more common among younger and female patients, there is a growing demand for cosmetic outcomes with high expectations. Traditional surgical incisions can leave scars on the neck, which can cause significant psychological burden, especially for patients with a tendency to form scars [2, 23]. In all three groups, patients recovered well after surgery, and the overall satisfaction rate at 3 months post-surgery was 100%. During the lateral approach, surgical access to the thyroid gland is achieved through the muscle interval between the clavicular head of the sternocleidomastoid muscle and the sternal head. This technique avoids the need for detaching a neck skin flap, eliminating the requirement for suturing the neck white line post-surgery. Furthermore, this approach minimizes disruption to the anterior neck muscles, resulting in reduced discomfort in the anterior neck region and no sensation of skin-trachea linkage during swallowing. As a result, the swallowing function of the anterior neck area is preserved [14, 24, 25]. There are no scars on the neck after surgery, and scars under the subclavicular and in the axilla can be completely covered, which can relieve patients' psychological burden and achieve excellent cosmetic outcomes.

## 5 Conclusion

In conclusion, these results indirectly demonstrate that the three lateral approaches are equally feasible and effective in the treatment of unilateral thyroid carcinoma. Compared to conventional incisions, the lateral approaches provide satisfactory cosmetic outcomes. With the gradual increase in the length of the approach, a step-by-step approach and the accumulation of early anatomical experience, the success rate for beginners can be significantly increased, and most surgeons with traditional thyroidectomy experience can easily perform this technique with a shorter learning curve. Additionally, the lateral approaches directly access the area of the recurrent laryngeal nerve and the parathyroid gland, ensuring surgical effectiveness. However, this study has some limitations. Firstly, this incision is only suitable for unilateral thyroid surgery. Secondly, for patients with abundant neck fat or developed muscles, the surgical operation space exposure may be relatively poor, leading to increased surgical difficulty. Finally, the sample size of this study is relatively small, and the follow-up time is short. The exploration of this surgical approach is still a long way to go, and it is hoped that through continuous exploration and conducting large-scale, multicenter studies, it can achieve more widespread clinical applications.

## Supporting information

**S1 Data.**
(ZIP)

## Author Contributions

**Data curation:** Yong-Xue Gu, Xi-Hao Ni.

**Investigation:** Wei-Tao Wang.

**Methodology:** Ying Lei.

**Resources:** Jing-Yi Ma.

**Software:** Jing-Yi Ma.

**Supervision:** Yong-Xue Gu, Xiao-Juan Men.

**Writing – original draft:** Ran An.

**Writing – review & editing:** Yong-Xue Gu, Chang-Liang Wang.

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
