## [Decision Letter · Decision Letter 0]

16 Nov 2023

PONE-D-23-28247The feasibility and clinical significance of lateral approach thyroidectomyPLOS ONE

Dear Dr. Wang,

Thank you for submitting your manuscript to PLOS ONE. After careful consideration, we feel that it has merit but does not fully meet PLOS ONE’s publication criteria as it currently stands. Therefore, we invite you to submit a revised version of the manuscript that addresses the points raised during the review process.

We look forward to receiving your revised manuscript.

Kind regards,

Antonino Maniaci

Academic Editor

PLOS ONE

3. Please remove your figures from within your manuscript file, leaving only the individual TIFF/EPS image files, uploaded separately. These will be automatically included in the reviewers’ PDF.

4. We note that Figure 1 in your submission contain copyrighted images. All PLOS content is published under the Creative Commons Attribution License (CC BY 4.0), which means that the manuscript, images, and Supporting Information files will be freely available online, and any third party is permitted to access, download, copy, distribute, and use these materials in any way, even commercially, with proper attribution. For more information, see our copyright guidelines: http://journals.plos.org/plosone/s/licenses-and-copyright.

Additional Editor Comments:

Please perform all the revisions required.

Reviewers' comments:

Reviewer's Responses to Questions

**Comments to the Author**

1. Is the manuscript technically sound, and do the data support the conclusions?

Reviewer #1: Partly

Reviewer #2: Partly

2. Has the statistical analysis been performed appropriately and rigorously? 

Reviewer #1: Yes

Reviewer #2: Yes

3. Have the authors made all data underlying the findings in their manuscript fully available?

Reviewer #1: Yes

Reviewer #2: Yes

4. Is the manuscript presented in an intelligible fashion and written in standard English?

Reviewer #1: Yes

Reviewer #2: Yes

5. Review Comments to the Author

Reviewer #1: I read with interest the manuscript by An et al. on the feasibility and clinical significance of lateral approach thyroidectomy. I have some comments to report to improve the manuscript:

- Authors should report ethical committee approval with code and date of approval.

- Why authors did not compare the lateral approach with the standard one? Please clarify.

- "Of the 52 patients, 8 were male and 44 were female, with an age range of 31-50 years and a mean age of (41.75±9.27) years." This information should be provided in the results section.

- Where is Weifang People's Hospital located? Please specify.

- Please include the outcomes studied under the subparagraph "outcomes" in the methods section.

- In all the tables provided, rows and columns must be inverted, to increase readability.

- In table 3, 4 and 5 there are some characters in chinese. please remove them.

- Please report a paragraph with the limitations of the study.

Reviewer #2: Introduction:

- Provide more background on the increasing incidence of thyroid cancer and demand for approaches that optimize cosmetic outcomes. Cite doi:10.1007/s00464-010-1341-2.

- Discuss limitations of conventional thyroidectomy incisions in meeting cosmetic expectations, especially in younger female patients. doi:10.1007/s00464-009-0347-0.

- Introduce various alternative surgical approaches for thyroid cancer, like transoral, retroauricular etc. and their pros and cons.

- Provide more details on the lateral approaches and explain how their incremental dissection could allow sequential accumulation of anatomical experience but increased risk of vocal cord paralysis. cite doi:10.2217/fon-2019-0053

- Clearly state the rationale and objectives of comparing feasibility and outcomes of the three lateral approaches.

Materials and Methods:

- Expand on inclusion and exclusion criteria for patient selection. Provide the sample size for each surgical approach group.

- Give more details about surgical equipment used. Specify any specialized instruments.

- Provide more information about general anesthesia protocols and patient positioning for each approach.

- Explain the step-by-step surgical techniques for each lateral approach, highlighting key anatomical landmarks.

- List all parameters that were observed and recorded during surgery and follow-up.

Results:

- Present relevant demographic and clinical characteristics of the patients in a table, with statistical comparisons between groups.

- Provide exact p-values and indicators of variance for all numerical outcome comparisons discussed between the groups.

- Include tables to summarize operative parameters, complications, pain scores, and satisfaction rates, with statistical significance marked.

- Comment on any notable patterns and variations between the three groups for the different outcome measures.

Discussion:

- Compare your cosmetic and other outcome results with prior studies on lateral and other minimal access thyroidectomy techniques. cite doi:10.1007/s00464-009-0820-9.

- Discuss possible factors contributing to differences in operation time between the approaches.

- Interpret the low complication rates in context of meticulous dissection techniques used.

- Suggest future research directions, like randomized controlled trials, long-term cosmetic follow-up, application for bilateral thyroidectomy etc.

6. PLOS authors have the option to publish the peer review history of their article (what does this mean?). If published, this will include your full peer review and any attached files.

Reviewer #1: No

Reviewer #2: No

---

## [Author Response · Author response to Decision Letter 0]

21 Dec 2023

Dear Editor and Reviewers,

Thank you very much for taking the time to review this manuscript amid your busy schedule. I sincerely appreciate all your comments and suggestions. Below, you will find my itemized responses, and in the resubmitted files, you can locate my revisions/corrections. We have submitted two versions of the manuscript, one with marked changes and the other without markings. Thanks again!

Responses to Editor

1.Please ensure that your manuscript meets PLOS ONE's style requirements, including those for file naming.

Author response: Ok, thanks!

2.We note that the grant information you provided in the ‘Funding Information’ and ‘Financial Disclosure’ sections do not match.

Author response: Ok, thanks!

3.Please remove your figures from within your manuscript file, leaving only the individual TIFF/EPS image files, uploaded separately. These will be automatically included in the reviewers’ PDF.

Author response: Ok, thanks!

4.We note that Figure 1 in your submission contain copyrighted images. All PLOS content is published under the Creative Commons Attribution License (CC BY 4.0), which means that the manuscript, images, and Supporting Information files will be freely available online, and any third party is permitted to access, download, copy, distribute, and use these materials in any way, even commercially, with proper attribution. For more information, see our copyright guidelines: http://journals.plos.org/plosone/s/licenses-and-copyright.

Author response: We change and modify the figure.

5.Additionally, I would like to propose an adjustment to the authorship order. Considering Yong-Xue Gu's substantial contributions during the early stages of the project and her significant efforts in revising the manuscript, we kindly request your approval to designate Yong-Xue Gu and Ran An as co-first authors. We believe that this change accurately reflects their respective contributions to the work.

Responses to Reviewer 1 

Reviewer point #1: Authors should report ethical committee approval with code and date of approval.

Author response #1: We added ethics committee approval with code 2023YX134 and an approval date of November 29, 2023. (Line 86-87)

Reviewer point #2: Why authors did not compare the lateral approach with the standard one? Please clarify.

Author response #2: When compared with the standard approach, the lateral approach undoubtedly results in longer procedural time and superior aesthetic outcomes. Therefore, the inclusion of such a comparison might make the study too broad, making it challenging to emphasize key points. The focus of this research is to highlight three pathways that progressively lengthen, step by step, gradually accumulating anatomical experience, with a short learning curve for clinicians. We chose to concentrate on a specific aspect to provide a more in-depth analysis. 

Reviewer point #3:  "Of the 52 patients, 8 were male and 44 were female, with an age range of 31-50 years and a mean age of (41.75±9.27) years." This information should be provided in the results section.

Author response #3: This information is provided in the results section. (Line 120-126、Line 183-184)

Reviewer point #4: Where is Weifang People's Hospital located? Please specify.

Author response #4: The location of Weifang People's Hospital is provided in the Correspondence section of the manuscript. (Line 11-13)

Reviewer point #5: Please include the outcomes studied under the subparagraph "outcomes" in the methods section.

Author response #5: The results section contains the full study results, and the methods section contains the observational Indicators related to the results.

Reviewer point #6: In all the tables provided, rows and columns must be inverted, to increase readability.

Author response #6: We have inversion is made for the rows and columns. 

Reviewer point #7: In table 3, 4 and 5 there are some characters in chinese. please remove them.

Author response #7: We remove the Chinese characters. 

Reviewer point #8: Please report a paragraph with the limitations of the study.

Author response #8: We add the content about the limitations of this study. (Line 388-395)

Responses to Reviewer 2

Introduction:

Reviewer point #1: Provide more background on the increasing incidence of thyroid cancer and demand for approaches that optimize cosmetic outcomes. Cite doi:10.1007/s00464-010-1341-2.

Author response #1: Please refer to lines 53-55 and Reference 2 for details.

Reviewer point #2: Discuss limitations of conventional thyroidectomy incisions in meeting cosmetic expectations, especially in younger female patients. doi:10.1007/s00464-009-0347-0.

Author response #2: Please refer to lines 358-362 and Reference 23 for details.

Reviewer point #3: Introduce various alternative surgical approaches for thyroid cancer, like transoral, retroauricular etc. and their pros and cons.

Author response #3: Please refer to lines 241-265 in the Discussion section.

Reviewer point #4: Provide more details on the lateral approaches and explain how their incremental dissection could allow sequential accumulation of anatomical experience but increased risk of vocal cord paralysis. cite doi:10.2217/fon-2019-0053

Author response #4: Please refer to lines 353-354 and Reference 22 for details.

Reviewer point #5: Clearly state the rationale and objectives of comparing feasibility and outcomes of the three lateral approaches.

Author response #5: Please refer to lines 60-75 in the manuscript.

Materials and Methods:

Reviewer point #1: Expand on inclusion and exclusion criteria for patient selection. Provide the sample size for each surgical approach group.

Author response #1: We have already selected appropriate inclusion and exclusion criteria and provided the sample sizes for each surgical approach group, which are 31, 13, and 8, respectively. (Line 88-90、Line 92-101)

Reviewer point #2: Give more details about surgical equipment used. Specify any specialized instruments.

Author response #2: More details have been provided about the surgical equipment used. (Line 105-114)

Reviewer point #3: Provide more information about general anesthesia protocols and patient positioning for each approach.

Author response #3: Please refer to lines 116-121 in the manuscript. 

Reviewer point #4: Explain the step-by-step surgical techniques for each lateral approach, highlighting key anatomical landmarks.

Author response #4: We have made modifications and additions, please refer to lines 122-149 for details. 

Reviewer point #5: List all parameters that were observed and recorded during surgery and follow-up.

Author response #5: Please refer to the Results section for details. 

Results:

Reviewer point #1: Present relevant demographic and clinical characteristics of the patients in a table, with statistical comparisons between groups.

Author response #1: Please refer to Section 2.1 for the comparison of patients' baseline characteristics. 

Reviewer point #2: Provide exact p-values and indicators of variance for all numerical outcome comparisons discussed between the groups.

Author response #2: Please refer to the Results section. 

Reviewer point #3: Include tables to summarize operative parameters, complications, pain scores, and satisfaction rates, with statistical significance marked.

Author response #3: Please refer to Section 2.2 -2.5.

Reviewer point #4: Comment on any notable patterns and variations between the three groups for the different outcome measures.

Author response #4: There were no significant differences among the three approach groups in terms of patient characteristics, number of central lymph node dissections, intraoperative blood loss, postoperative drainage volume, duration of drainage tube placement, length of hospital stay, postoperative pain, satisfaction, and complications. However, the operation time was longest in the subclavicular approach group, followed by the axillary approach group, and shortest in the supraclavicular approach group. The total hospitalization cost was highest in the axillary approach group, followed by the subclavicular approach group, and lowest in the supraclavicular approach group. Refer to the Results section. 

Discussion:

Reviewer point #1: Compare your cosmetic and other outcome results with prior studies on lateral and other minimal access thyroidectomy techniques. cite doi:10.1007/s00464-009-0820-9.

Author response #1: Please refer to lines 368-372 and Reference 25 for details.

Reviewer point #2: Discuss possible factors contributing to differences in operation time between the approaches.

Author response #2: Please refer to the Discussion section 3.2 for the analysis of operation time. (Line 317-340)

Reviewer point #3: Interpret the low complication rates in context of meticulous dissection techniques used.

Author response #3: Please refer to lines 248-252 in the Discussion section.

Reviewer point #4: Suggest future research directions, like randomized controlled trials, long-term cosmetic follow-up, application for bilateral thyroidectomy etc.

Author response #4: The study has certain limitations, and future research should focus on large-sample, multicenter studies to achieve broader clinical applicability. (Line 388-395)

---

## [Decision Letter · Decision Letter 1]

7 Jan 2024

PONE-D-23-28247R1The feasibility and clinical significance of lateral approach thyroidectomyPLOS ONE

Dear Dr. Wang,

Thank you for submitting your manuscript to PLOS ONE. After careful consideration, we feel that it has merit but does not fully meet PLOS ONE’s publication criteria as it currently stands. Therefore, we invite you to submit a revised version of the manuscript that addresses the points raised during the review process.

We look forward to receiving your revised manuscript.

Kind regards,

Antonino Maniaci

Academic Editor

PLOS ONE

Journal Requirements:

Additional Editor Comments:

Please revise the paper according to the suggestions. Best regards.

Reviewers' comments:

Reviewer's Responses to Questions

**Comments to the Author**

1. If the authors have adequately addressed your comments raised in a previous round of review and you feel that this manuscript is now acceptable for publication, you may indicate that here to bypass the “Comments to the Author” section, enter your conflict of interest statement in the “Confidential to Editor” section, and submit your "Accept" recommendation.

Reviewer #1: (No Response)

Reviewer #2: (No Response)

Reviewer #3: All comments have been addressed

2. Is the manuscript technically sound, and do the data support the conclusions?

Reviewer #1: (No Response)

Reviewer #2: Yes

Reviewer #3: Yes

3. Has the statistical analysis been performed appropriately and rigorously? 

Reviewer #1: (No Response)

Reviewer #2: Yes

Reviewer #3: Yes

4. Have the authors made all data underlying the findings in their manuscript fully available?

Reviewer #1: (No Response)

Reviewer #2: Yes

Reviewer #3: Yes

5. Is the manuscript presented in an intelligible fashion and written in standard English?

Reviewer #1: (No Response)

Reviewer #2: Yes

Reviewer #3: Yes

6. Review Comments to the Author

Reviewer #1: The authors successfully addressed all the comments provided. I believe the manuscript is now ready for acceptance in its present form.

Reviewer #2: the paper is intersting but needs minor structural revisions:

Methods:

- Provide more details on patient selection criteria, such as inclusion/exclusion criteria related to tumor stage, disease status, other medical conditions, etc.

- Describe surgical techniques in more granular steps rather than just approach name. Identify key steps like nerve monitoring, hemostasis methods.

- Specify equipment/instruments used like endoscopic system, energy devices, monitoring systems, operation table configuration.

- Explain outcome measures and how they were assessed - define variables clearly like blood loss (intra-op vs total), complications (grading system), satisfaction scales.

- Describe statistical analysis in more detail - what tests were used for which outcomes based on data type/distribution. Specify significance level.

Results:

- Provide summary stats for sample characteristics like mean/median values for age, tumor size rather than just ranges.

- Use tables to present results clearly for easy comparison between groups.

- Consider graphical displays like boxplots for non-normally distributed data.

- Conduct subgroup/stratified analysis where appropriate to identify effect modifiers.

- Report both numerical values and percentages for categorical variables like complications.

- Interpret statistically significant vs non-significant findings in the discussion.

- Highlight key findings and discuss implications for clinical practice and further research.

Reviewer #3: I think authors have addressed the questions raised by the previous reviewers. I have no further comments.

7. PLOS authors have the option to publish the peer review history of their article (what does this mean?). If published, this will include your full peer review and any attached files.

Reviewer #1: No

Reviewer #2: **Yes: **Salvatore

Reviewer #3: No

---

## [Author Response · Author response to Decision Letter 1]

25 Jan 2024

Dear Editor and Reviewers,

Thank you very much for taking the time to review this manuscript amid your busy schedule. I sincerely appreciate all your comments and suggestions. Below, you will find my itemized responses, and in the resubmitted files, you can locate my revisions/corrections. We have submitted two versions of the manuscript, one with marked changes and the other without markings. Thanks again!

Responses to Reviewer 1 

Reviewer point #1: The authors successfully addressed all the comments provided. I believe the manuscript is now ready for acceptance in its present form.

Author response #1: Appreciate your time and valuable input.

Responses to Reviewer 2

Methods:

Reviewer point #1: Provide more details on patient selection criteria, such as inclusion/exclusion criteria related to tumor stage, disease status, other medical conditions, etc.

Author response #1: The inclusion standards remain unchanged, while exclusion criteria have been added.（Line99-104）

Reviewer point #2: Describe surgical techniques in more granular steps rather than just approach name. Identify key steps like nerve monitoring, hemostasis methods.

Author response #2: Detailed steps for three lateral approach surgical procedures are described.（Line119-152）

Reviewer point #3: Specify equipment/instruments used like endoscopic system, energy devices, monitoring systems, operation table configuration.

Author response #3: The models and manufacturers of the instruments and equipment have been provided. （Line108-117）

Reviewer point #4: Explain outcome measures and how they were assessed - define variables clearly like blood loss (intra-op vs total), complications (grading system), satisfaction scales.

Author response #4: We have explicitly defined variables in the original text. For example, intraoperative blood loss is defined as the volume of bleeding during surgery, and postoperative drainage is defined as the amount of drainage following the operation. All observed complications were transient, and no severe complications occurred; therefore, no grading was performed. The transient nature is explicitly stated in the original text. Satisfaction comparison was assessed through postoperative follow-up scoring on a 5-point scale.（In the results section）

Reviewer point #5: Describe statistical analysis in more detail - what tests were used for which outcomes based on data type/distribution. Specify significance level.

Author response #5: Continuous data were presented as mean ± standard deviation (x±s). Analysis of variance (ANOVA) was used for normally distributed data, and non-parametric tests were used for data with non-uniform variances. Categorical data were presented as frequency (n) and percentage (%), and the chi-square test was used for intergroup comparisons. A p-value of less than 0.05 was considered statistically significant.（Line175-181）

Results:

Reviewer point #1: Provide summary stats for sample characteristics like mean/median values for age, tumor size rather than just ranges.

Author response #1: In Section 2.1.（Line186-196）

Reviewer point #2: Use tables to present results clearly for easy comparison between groups.

Author response #2: See Results section.

Reviewer point #3: Consider graphical displays like boxplots for non-normally distributed data.

Author response #3: The box plots have been added to the manuscript. （Line208-212、216-217）

Reviewer point #4: Conduct subgroup/stratified analysis where appropriate to identify effect modifiers.

Author response #4: Considering our research objectives, we believe that subgroup/stratified analysis would not be meaningful.

Reviewer point #5: Report both numerical values and percentages for categorical variables like complications.

Author response #5: Refer to Sections 2.3 and 2.5.

Reviewer point #6: Interpret statistically significant vs non-significant findings in the discussion.

Author response #6: Refer to the discussion section 3.1-3.4 for the analysis of the results section.

Reviewer point #7: Highlight key findings and discuss implications for clinical practice and further research.

Author response #7: Refer to lines 391-408.

Responses to Reviewer 3

Reviewer point #1: I think authors have addressed the questions raised by the previous reviewers. I have no further comments.

Author response #1: Appreciate your time and valuable input.

---

## [Decision Letter · Decision Letter 2]

1 Mar 2024

The feasibility and clinical significance of lateral approach thyroidectomy

PONE-D-23-28247R2

Dear Authors,

We’re pleased to inform you that your manuscript has been judged scientifically suitable for publication and will be formally accepted for publication once it meets all outstanding technical requirements.

Kind regards,

Antonino Maniaci

Academic Editor

PLOS ONE

Additional Editor Comments (optional):

Dear authors, it's a pleasure to propose acceptation of the paper. Bests

Reviewers' comments:

Reviewer's Responses to Questions

Reviewer #2: All comments have been addressed

2. Is the manuscript technically sound, and do the data support the conclusions?

Reviewer #2: Yes

3. Has the statistical analysis been performed appropriately and rigorously? 

Reviewer #2: Yes

4. Have the authors made all data underlying the findings in their manuscript fully available?

Reviewer #2: Yes

5. Is the manuscript presented in an intelligible fashion and written in standard English?

Reviewer #2: Yes

6. Review Comments to the Author

Reviewer #2: all the revisions were performed, the paper is interestingly improved and now can be accepted.

Bests

7. PLOS authors have the option to publish the peer review history of their article (what does this mean?). If published, this will include your full peer review and any attached files.

Reviewer #2: **Yes: **Salvatore

---

## [Editor Report · Acceptance letter]

7 Mar 2024

PONE-D-23-28247R2 

PLOS ONE

Dear Dr. Wang, 

I'm pleased to inform you that your manuscript has been deemed suitable for publication in PLOS ONE. Congratulations! Your manuscript is now being handed over to our production team.

Kind regards, 

on behalf of

Prof. Antonino Maniaci 

Academic Editor

PLOS ONE